

# The Impact of Neglecting Ice Phase on Cloud Optical Depth Retrievals from AERONET Cloud Mode Observations

Jonathan K. P. Shonk[1], Jui-Yuan Christine Chiu[2], Alexander Marshak[3], David M. Giles[3, 4], Chiung-Huei
Huang[5], Gerald G. Mace[6], Sally Benson[6], Ilya Slutsker[3, 4] and Brent N. Holben[3]

[1]National Centre for Atmospheric Science, Department of Meteorology, University of Reading, Reading, UK
[2]Department of Atmospheric Science, Colorado State University, Fort Collins, CO, 80523, USA
[3]NASA/Goddard Space Flight Center, Greenbelt, Maryland, USA
[4]Science Systems and Applications, Inc., Lanham, Maryland, USA
[5]Center for Environmental Monitoring and Technology, National Central University, Taoyuan, Taiwan
[6]Department of Atmospheric Sciences, University of Utah, Salt Lake City, Utah, USA

*Correspondence to*: Jonathan K. P. Shonk (j.k.p.shonk@reading.ac.uk)

**Abstract.** Cloud optical depth remains a difficult variable to represent in climate models, and hence there is a need for high-
quality observations of cloud optical depth from locations around the world. Such observations could be readily obtained from
Aerosol Robotic Network (AERONET) radiometers using a two-wavelength retrieval method. However, the method requires
an assumption that all of the cloud in a profile is liquid, and this has the potential to introduce errors into long-term statistics
of retrieved optical depth. Using a set of idealised cloud profiles, we find that the fractional error in retrieved optical depth is
a linear function of the fraction of the optical depth that is due to the presence of ice cloud ("ice fraction"), with a magnitude
of order 55% to 70% for clouds that are entirely ice. We derive a simple linear equation that could potentially be used as a
correction at AERONET sites where ice fraction can be independently estimated.

The greatest contribution to error statistics arises from optically thick cloud that is either mostly or entirely ice. Using this
linear equation, we estimate the magnitude of the error for a set of cloud profiles measured at five sites of the Atmospheric
Radiation Measurement programme. Instances of such clouds are not frequent, with less than 15% of cloud profiles at each
location showing an error of greater than 10. However, differences in the frequency of such clouds from one location to another
affect the magnitude of the overall mean error, with sites dominated by deep tropical convection and thick frontal mixed-phase
cloud showing greater errors than sites where deep clouds are less frequent. The mean optical depth error at the five locations
spans the range 2.5 to 5.5, which we show to be small enough to allow calculation of top-of-atmosphere flux to within 10%,
and surface flux to about 15%.



# 1 Introduction

Clouds are a crucial part of the climate system, yet present many great challenges to climate science (Randall *et al*, 2007; Boucher *et al*, 2013). Part of the challenge is the representation of small-scale structure and processes in climate models, which requires parameterisation (Pincus *et al*, 2003; Shonk and Hogan, 2010). Despite recent progress, however, models still struggle

to represent cloud optical properties (Bender *et al*, 2006; Lauer and Hamilton, 2013; Klein *et al*, 2013; Calisto *et al*, 2014). Cloud optical depth is an important variable in cloud models and, to make further advances in our understanding of cloud processes, we need global observations of this quantity at high temporal and spatial resolution.

The standard approach to measure cloud optical depth is to retrieve it remotely from measurements of reflectance, radiance or

irradiance in multiple spectral bands. Following this principle, various methods have been developed to retrieve cloud optical depth from satellite measurements (for example, Arking and Childs, 1985; Nakajima and King, 1990; Platnick *et al*, 2001; Cooper *et al*, 2007) and ground-based instruments (Marshak *et al*, 2000, 2004; Barker and Marshak, 2001; Chiu *et al*, 2006). The need for global observations is best met by satellites, which are capable of providing routine cloud optical depth retrievals all around the world. However, on account of their large pixel size, they struggle to provide the high temporal and spatial

resolution required to investigate cloud processes. The underlying surface adds to the complexity of variability in the optical properties, and broken clouds and subpixel clouds increase the chance of errors and biases (Stephens and Kummerow, 2007). Using ground-based observations eliminates many of these issues. The proximity of clouds to the ground (much closer than a satellite orbit) means that a radiometer can achieve much smaller pixel sizes for the same viewing angle, allowing much higher temporal and spatial resolution, and reducing the incidences of cloud edge.

A disadvantage of using ground-based observations is the lack of global coverage. We are limited to the small number of locations around the world where routine cloud optical depth observations are made: until recently, sites of the Atmospheric Radiation Measurement (ARM) Programme (Stokes and Schwartz, 1994) and the sites of the Aerosols, Clouds and Trace Gases Research Infrastructure (ACTRIS) network that were formerly part of Cloudnet (Illingworth *et al*, 2007). But Chiu *et*

*al* (2010) noted that radiometers distributed throughout the world as part of the AERONET project (Holben *et al*, 1998) could provide a readily available source of cloud optical depth observations and hence provide greater global coverage. When the sun is not obscured by cloud, these radiometers are in "aerosol mode" and make regular measurements of aerosol properties. When the sun is obscured, however, aerosol measurements are not possible and the radiometer becomes idle. Marshak *et al* (2004) proposed that the "down-time" when the aerosol measurements are not possible could be used to observe cloud

properties ("cloud mode") via measurements of zenith radiance.

Cloud optical depth retrievals are made using the method proposed by Chiu *et al* (2010). It is based on that of Marshak et al (2004), and uses radiances measured at two wavelengths (440 nm and 870 nm; one visible, one infra-red) to retrieve cloud



optical depth and cloud fraction. At these wavelengths, the radiative properties of the clouds are similar, but the albedo of the surface, here assumed to be green vegetation, is very different. Using this method, AERONET "cloud mode" optical depth retrievals have now been made routinely at a number of sites around the world for several years, and are beginning to appear in published studies. An evaluation of data from one AERONET site in Cuba was made by Barja *et al* (2012). Antón *et al*

(2012) used cloud mode data in a study into the effects of cloud optical depth on the transmission of ultra-violet radiation; Li *et al* (2018) used it to investigate seasonal and spatial distributions of cloud optical depth across China alongside satellite optical depth retrievals from MODIS (the Moderate Resolution Imaging Spectroradiometer; Platnick *et al*, 2003). An AERONET radiometer was also taken aboard a ship to probe the properties of boundary layer cloud in the north-eastern tropical Pacific (Painemal *et al*, 2017).

An extension to the retrieval method by Chiu *et al* (2012) included a third wavelength in the process (1640 nm), which allows a retrieval of cloud droplet effective radius to be obtained alongside cloud optical depth and cloud fraction. Effective radius retrievals tend to be very sensitive to uncertainty in surface albedo and radiance measurements, so Chiu *et al* (2012) suggested performing the retrieval 40 times with perturbations to surface albedo and the measured radiance, thereby providing mean

values of the retrieved values and an estimate of the uncertainty in these retrievals. This method was used in the study of Painemal *et al* (2017), although the standard retrievals available on the AERONET website use the two-wavelength method of Chiu *et al* (2010).

However, neither of these retrieval methods are capable of retrieving cloud phase, so an assumption is made. Given the

tendency for the liquid component of a cloudy profile to be substantially optically thicker than the ice component, it is assumed that the entirety of the retrieved cloud optical depth value is due to the presence of liquid cloud. This "warm cloud assumption" has the potential, therefore, to introduce an error into cloud optical depth retrievals in any case where a cloudy profile contains ice cloud, which could cause problems in studies that analyse long-term statistics of cloud optical depth.

The objectives of this study are to (1) investigate the magnitude and sign of the retrieval error due to the warm cloud assumption, (2) ascertain whether it is large enough to drastically affect the statistics of long-term optical depth retrievals and, if necessary, (3) discover whether a simple correction method could be used to account for the error. The next section of this paper describes the Chiu *et al* (2010) retrieval method in more detail and provides a first estimate of the sign and magnitude of the error. In Section 3, we examine the relationship of the error with both total cloud optical depth and how the optical depth

is partitioned between ice and liquid components by performing retrievals on a set of idealised cloud profiles. From these results, we propose a simple linear correction equation that could be employed in AERONET locations where ice fraction can be independently determined. In Section 4, we investigate the potential magnitude of the error in real clouds measured at five ARM sites using retrieval methods described by Mace *et al* (2006). We then summarise the study in Section 5.



## 2 Two-channel retrieval method

Retrievals throughout this study are performed using the two-channel method described by Chiu *et al* (2010). The method

begins with a set of look-up tables, which contain the radiance that would be observed at the surface under a cloudy profile for

a range of different cloud optical depths, solar zenith angles and values of droplet effective radius. Using the Discrete Ordinate

Method for Radiative Transfer radiation code (DISORT; Stamnes *et al*, 1988), we calculate a set of tables for each of the two

wavelength channels, 440 nm and 870 nm. The surface albedo in the two channels is set to 0.05 and 0.35 respectively (typical

albedo values over a green vegetated surface as reported by Chiu *et al*, 2010). The scattering properties applied to DISORT

for all look-up table calculations are those of liquid water droplets.

A pair of measured radiances at the two wavelengths is fed into the retrieval algorithm along with an assumed liquid effective

radius (taken to be 8 µm throughout this study) and the known solar zenith angle at that time. From the look-up tables, the

algorithm then searches for values of optical depth and cloud fraction that produce the specified radiance at both wavelengths.

To estimate the uncertainty on the retrieval, we follow part of the method of Chiu *et al* (2012) and perform 40 such calculations,

each one with a random perturbation applied to both the surface albedos and the observed radiances to represent uncertainty

in their measurement. The output retrieved optical depth and cloud fraction therefore consist of a mean value and an indication

of uncertainty.

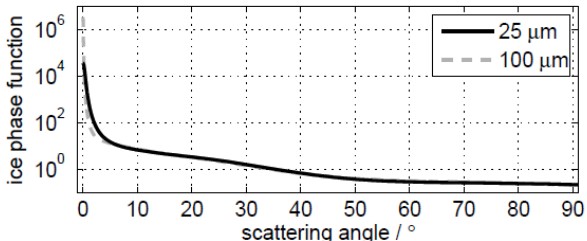

**Figure 1. Ice phase functions used in this study, originally designed for use in cloud retrievals from MODIS. Phase functions are**
**shown for the forward scattering direction at wavelength 465 nm, for two ice particle effective diameters (see legend).**

To make an initial estimate of the sign and magnitude of the "warm cloud error", we use DISORT to calculate a few look-up

tables using scattering properties of ice particles and compare them with the corresponding look-up tables calculated using the

properties of liquid droplets. We use a set of ice crystal phase functions for a randomly aligned distribution of rough-surfaced

ice crystals, consisting of a mixture of shapes (a "general habit mixture"), retrieved from www.ssec.wisc.edu/ice_models/.

These phase functions were calculated alongside other single-scattering properties from field campaign data by Baum *et al*

(2011, 2014). Their calculated properties are designed for use with radiative transfer calculations that allow retrieval of optical

properties from satellites, with a different set of properties for each satellite platform to allow consistent retrieval. Given the





availability of phase functions near the two wavelengths used in AERONET cloud optical depth retrievals, we select the phase functions designed for MODIS. Figure 1 shows the ice phase functions at wavelength 465 nm for particles with effective diameters of 25 µm and 100 µm (the range of effective diameters that we consider in this study). The corresponding phase functions at 855 nm are similar.

Figure 2 compares the radiances that would be observed at the surface at the respective visible wavelengths under a column of cloud that is either purely ice or purely liquid, for a prescribed solar zenith angle of 30°. For a given optical depth, the observed radiance for liquid clouds is always more than that for an ice cloud of the same optical depth over the entire range of effective sizes used in this study. This is because liquid droplets have a greater tendency to forward scatter than ice crystals, resulting

in a greater radiance at the surface for the same amount of extinction. For any profile whose true optical depth is in the branch of the curve on Figure 2 where the radiance is monotonically decreasing with increasing optical depth (that is, to the right of the maximum), the error in retrieved cloud optical depth will be positive. Consider an example: an observed radiance measurement is 0.4 W m$^{-2}$, and we assume that the cloud is liquid with an effective radius of 8 µm and has an optical depth greater than 10. From Figure 2, we would retrieve an optical depth of about 25. However, if all of the cloud is in fact rough

ice crystals with an effective diameter between 25 µm and 100 µm, the actual optical depth might only be between 16 and 17, implying a positive error of between 47% and 56%.

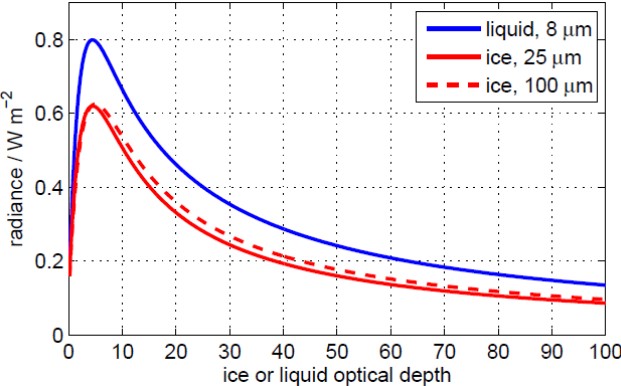

**Figure 2. Radiances extracted from the liquid (blue) and ice (red) look-up tables for a range of different optical depths, all calculated for a solar zenith angle of 30° and at the visible 440 nm wavelength over a surface of albedo 0.05. The numbers in the legend are**
**values of liquid effective radius and ice effective diameter.**

## 3   Errors in idealised cloud profiles

For a better understanding of the retrieval error, we use the two-channel retrieval method to obtain cloud optical depth for a set of idealised cloud profiles where the cloud optical depth is known. Each profile includes two cloudy layers: the top layer is filled with ice cloud and the bottom layer is filled with liquid cloud, both with a cloud fraction of one. The properties of

these cloud layers are varied in two ways. First, the total combined optical depth of the two layers is varied. Second, the





partitioning of this total column optical depth between the ice and liquid layer is varied. We define a variable called "ice fraction" – this is the fraction of the total column optical depth that is due to the presence of ice cloud. For each combination of optical depth and ice fraction, a full radiative transfer calculation is performed using DISORT to obtain the zenith radiance that would be detected at the surface by a vertically pointing radiometer, serving as the synthetic "observed" radiance. The

appropriate scattering properties are used for the liquid and ice layers. We fix liquid effective radius at 8 µm, and perform radiance calculations for ice effective diameters of 25 µm, 35 µm, 55 µm and 100 µm and for solar zenith angles of 10°, 30°, 50° and 70°, in both the 440 nm and 870 nm channels. Aerosol concentrations are set to zero.

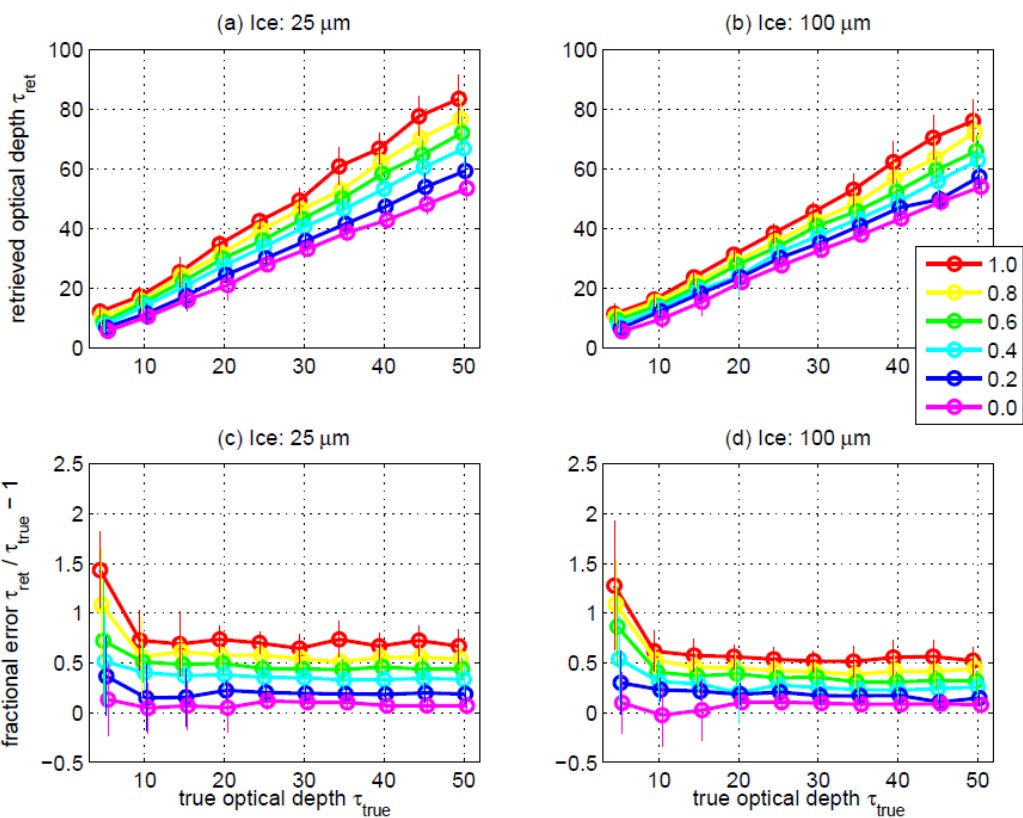

**Figure 3.** Retrieved optical depth ($\tau_{ret}$; top row), and retrieved optical depth as a fraction of prescribed ("true") optical depth
($\Delta\tau_{ret}/\tau_{true}$; bottom row) as a function of the true optical depth for the idealised cloud columns. Retrievals are made from DISORT radiance calculations with a liquid effective radius of 8 µm, a solar zenith angle of 30°, and two values of ice effective diameter (see panel headers). The lines and markers are coloured according to the ice fraction (see legend). The uncertainty in the retrieval, depicted here as the standard deviation in the retrievals across the 40 samples, is indicated by the vertical bars. Note that the markers and bars for each ice fraction value are slightly horizontally offset for clarity.

Retrievals of cloud optical depth are then made from the "observed" radiances under the assumption that all clouds are liquid. Figure 3 shows that the true optical depth is generally well matched by the retrieved optical depth for profiles that contain cloud that is entirely liquid (ice fraction equal to zero), while increasing ice fraction reduces the surface radiance for a given cloud optical depth and results in an increasingly positive error. Furthermore, at most optical depths shown here, the fractional





error in retrieved optical depth is largely independent of the true optical depth and increases linearly with increasing ice fraction. For clouds that are entirely ice (ice fraction equal to one), the fractional error reaches about 70% if the ice effective diameter is assumed to be 25 µm and about 55% if it is assumed to be 100 µm. The fractional error is also largely independent of solar zenith angle, remaining at about 70% when the ice effective diameter is fixed at 25 µm and the solar zenith angle is

varied (Figure 4).

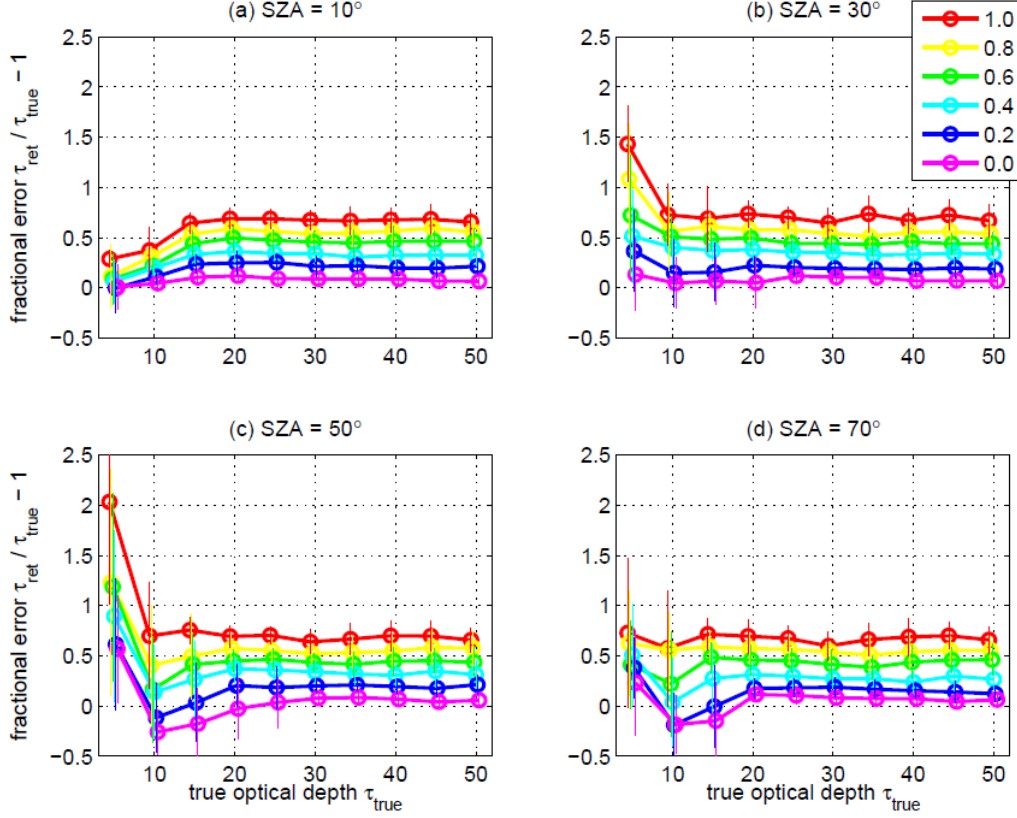

**Figure 4. Retrieved optical depth as a fraction of true optical depth ($\Delta\tau_{ret}/\tau_{true}$) as a function of the true optical depth in the idealised cloud columns. Retrievals are made from DISORT radiance calculations with a liquid effective radius of 8 µm, an ice effective diameter of 25 µm and four values of solar zenith angle (see panel headers). Lines and markers as described in Figure 3.**

At low optical depths (values below about 20), however, the relationship between fractional error and ice fraction becomes more complicated, with a dependence on both the true optical depth and the solar zenith angle. The range of low optical depths affected by this more complicated relationship is also dependent on solar zenith angle. A simple explanation for these two different "error regimes" arises from Figure 2, and how the shape of the curves change with changing solar zenith angle and ice fraction. At higher optical depths (the "linear regime"), the observed radiance decreases monotonically with increasing

optical depth. Changes to the ice fraction or solar zenith angle may change the nature of the curve, but do not change this monotonic behaviour. At lower optical depths (the "non-linear" regime), the change of shape does not just affect the gradients, but also the location of the maximum point of the curve, adding complicated non-linearity into the relationship.



Based on DISORT computations and the assumed ice cloud particle diameters above, the relationship between fractional error in retrieved optical depth $\Delta\tau/\tau_{\text{true}}$ and ice fraction $f$ in the "linear regime" could be quantified using a simple linear empirical equation of the form

$$\frac{\Delta\tau}{\tau_{\text{true}}} = (a \pm \Delta a)f + (b \pm \Delta b) \ , \tag{1}$$

where $a$ and $b$ are the regression coefficients, and $\Delta a$ and $\Delta b$ are the uncertainty in these coefficients. This regression is demonstrated in Figure 5, and yields coefficients of $a = 0.534$, $b = 0.067$ and $\Delta b = 0.052$. (The value of $\Delta a$ was found to be negligible and less than 0.001.) To ensure retrievals in the "non-linear regime" are excluded, this regression only includes profiles with a true optical depth of greater than 20. To include a measure of uncertainty in the size of the ice particles, we include retrievals for all four values of ice effective diameter. Given that the solar zenith angle is known for a retrieved profile,

it is perceivable to calculate regressions for each solar zenith angle separately and then add a solar zenith angle dependence to Equation 1. However, variations in the regression coefficients for different solar zenith angles were found to be small, so we include all four solar zenith angles in one single regression for simplicity.

A simple linear equation of this form could be used to correct the warm cloud error in AERONET optical depth retrievals if

an estimate of ice fraction could be independently derived at the AERONET site; for example, via separate retrievals of liquid and ice water paths from microwave radiometer and radar measurements respectively. While it is not capable of accounting for errors in the "non-linear regime" at low optical depths, it should provide reliable correction to all clouds with true optical depths of above 20 in the range of solar zenith angles considered here. In the optical depth range 10 to 20, applying the correction equation could lead to errors in some instances of high-sun or low-sun, although these are likely to be small (see

Figure 4). Below optical depths of 10, the reliability of the correction equation becomes questionable, as the fractional errors start to become large. However, this may not present a great problem when evaluating long-term cloud statistics from AERONET for two reasons. First, the absolute magnitude of errors in optically thinner cases is much smaller and hence far less of an issue than potential retrieval errors in optically thicker clouds. Second, at low cloud optical depths, the AERONET radiometers may be able to operate in aerosol mode and hence not be observing cloud. According to Giles *et al* (2019), the

maximum aerosol optical depth that can be measured by the radiometers is between about 5 and 7, depending on the radiometer type. In principle, further non-linear regression methods could be used to build a more complex correction equation that accounts for the error at low optical depths. In this study, however, we retain the simple linear regression presented above and accept its limitations.





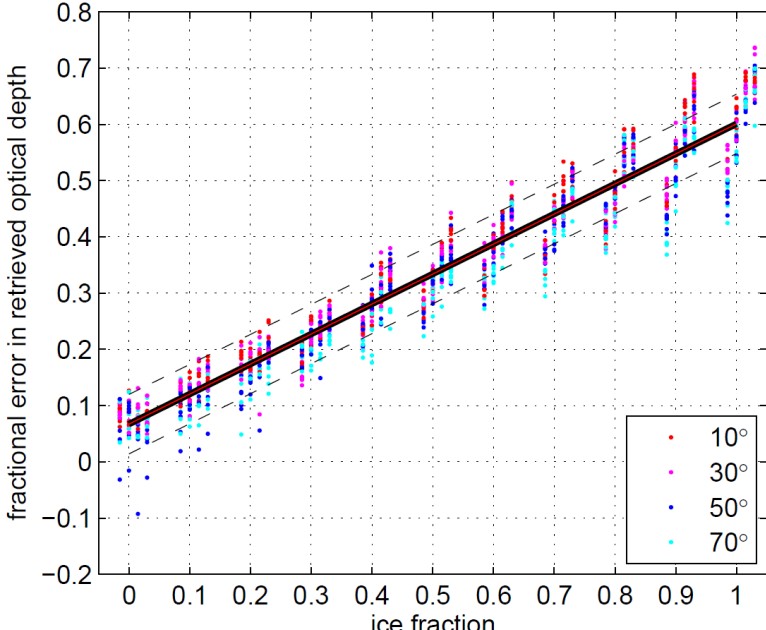

**Figure 5. Fractional error in the retrieved optical depth, calculated as $(\tau_{\text{ret}} - \tau_{\text{true}})/\tau_{\text{true}}$, for the idealised cloud columns as a function of the prescribed ice fraction (horizontal axis) and solar zenith angle (colours; see legend). The four columns of points around each 0.1 interval in ice fraction indicate the distributions of fractional error across the four values of ice effective diameter (25 μm, 35 μm, 55 μm and 100 μm from left to right). A linear fit through the points is shown (solid line), along with an estimate of its uncertainty (dashed lines).**

## 4 Statistics from real cloud profiles

For optically thick clouds with a high ice fraction, the error in retrieved optical depth can be large following Equation 1 (for a cloud that is entirely ice and has an optical depth of 50, for example, the error could be about 30). The question then follows as to how frequently such optically thick ice clouds occur at the location of the AERONET sites with "cloud mode" retrieval. The assumption that the liquid component of a cloudy profile tends to be optically thicker than the ice component, stated in Section 1, suggests that optically thick ice clouds may not be a frequent occurrence and hence only provide a small contribution to long-term statistics of cloud optical depth. In this section, we address this question by examining the distribution of optical depth and ice fraction in real clouds.

We therefore require a dataset that can provide independent values of ice and liquid components of optical depth at sites that contain AERONET radiometers that operate in cloud mode. We hence use cloud data measured at five ARM sites, using algorithms described by Mace *et al* (2006) and hereafter referred to as "ARM Mace" data. The methods of Mace *et al* (2006) derive a wealth of properties of an atmospheric profile using a combination of ground-based remote sensing techniques and





radiosonde soundings, and provide a series of cloud profiles averaged over 5-minute intervals with a vertical resolution of 90 m. Liquid water path is obtained from brightness temperatures measured in two wavelength channels by a microwave radiometer. Ice water content is determined from millimetre cloud radar measurements using two different methods, depending on whether the profile contains pure ice cloud or mixed-phase cloud. The former case uses one of a set of algorithms to

determine a distribution of ice water content from radar reflectivity and either Doppler velocity or longwave radiance at the surface; the latter uses a specially developed parameterisation that also uses reflectivity and Doppler velocity. Separate values of ice and liquid optical depth components are then calculated from the liquid water path and the vertically integrated ice water content, hence allowing an estimate of ice fraction.

**Table 1. A summary of cloud statistics across the five ARM sites discussed in this study. Profiles included in these statistics consist only of those from the ARM Mace dataset at times when an AERONET cloud mode retrieval would have been possible (see third and fourth paragraphs of Section 4 for criteria).**

|  | SGP | NSA | Manus | Nauru | Darwin |
|---|---|---|---|---|---|
| *Years of available data…* | 2005– | 2008– | 2005– | 2005– | 2005– |
|  | 2009 | 2010 | 2007 | 2007 | 2008 |
| *Number of profiles…* | 74,973 | 80,477 | 27,564 | 21,229 | 53,166 |
| *Percentage of profiles that contain…* |  |  |  |  |  |
| Liquid clouds | 26.5% | 17.0% | 16.9% | 37.1% | 29.3% |
| Mixed-phase clouds, $f < 0.5$ | 29.4% | 62.2% | 34.8% | 28.0% | 29.0% |
| Mixed-phase clouds, $f > 0.5$ | 10.8% | 14.6% | 14.4% | 4.8% | 4.1% |
| Mixed-phase clouds, all $f$ | 40.2% | 76.8% | 49.1% | 32.7% | 33.1% |
| Ice clouds | 33.3% | 6.2% | 34.0% | 30.2% | 37.6% |
| *Percentage of profiles with errors…* |  |  |  |  |  |
| Greater than 5 | 18.3% | 23.7% | 20.2% | 7.3% | 13.4% |
| Greater than 10 | 9.2% | 13.3% | 9.0% | 2.9% | 5.9% |
| Greater than 20 | 3.1% | 4.6% | 2.9% | 0.5% | 1.8% |
| **Mean error over all profiles** | **3.5** | **4.4** | **3.5** | **1.8** | **2.8** |

We fetch all available ARM Mace data from 2005 onwards at the Southern Great Plains site (SGP) in Oklahoma, the three

Tropical Western Pacific sites in Manus, Nauru and Darwin, and the North Slope of Alaska site (NSA) in Barrow. There are at least three years of data at each site, although the range of available years varies (see top part of Table 1). From this ARM Mace data, we extract profiles that could potentially be observed by an AERONET radiometer in cloud mode. We first remove all night-time profiles, and any profiles measured during periods of rainfall. Rainy profiles are indicated by the "precipitation flag" that is contained within the ARM Mace dataset; night-time profiles are identified by instances where the solar zenith





angle is greater than 90°. We also remove any profiles that contain a retrieved value of ice water content greater than 2 g m$^{-3}$, as such values cannot be considered reliable according to the ARM Mace documentation.

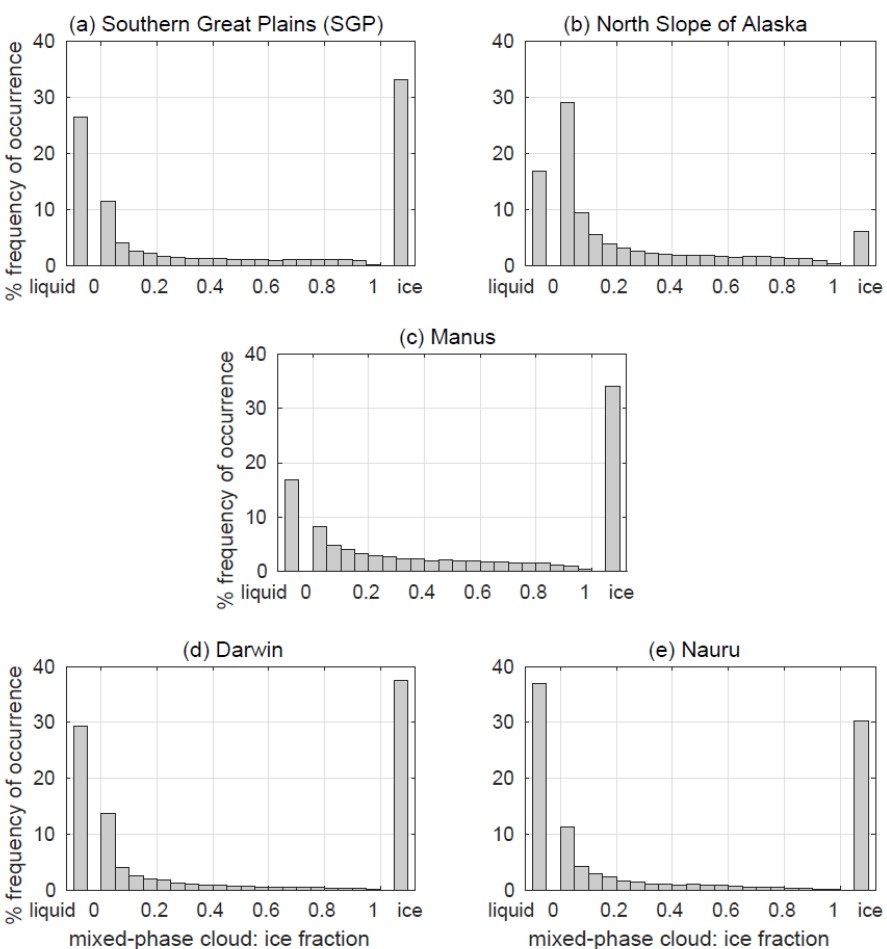

**Figure 6. Histograms of ice fraction for real clouds observed at five ARM sites. All available profiles in the period 2005 to 2010 are included for which an AERONET cloud mode retrieval would have been possible (see third and fourth paragraphs of Section 4 for conditions). The "liquid" and "ice" bars indicate the fraction of total profiles that contain purely liquid or ice; the "mixed-phase" bars indicate all other profiles, separated into bins of ice fraction. Data from the Mace *et al* (2006) dataset ("ARM Mace").**

Finally, we account for the upper limit of total optical depth that can be retrieved by the AERONET cloud mode algorithm by removing profiles that have a *retrieved* optical depth of greater than 100. Considering the ARM Mace optical depths to be the "*truth*", we use Equation 1 to simulate the AERONET cloud mode retrieval process, generating a set of "retrieved" optical depths. Any "retrieved" optical depths greater than 100 are excluded. The retrieval error for each profile is determined as the difference between the "true" and "retrieved" optical depth values.



It should be noted that this sample does not exclude profiles where the cloud optical depth is low, yet an AERONET aerosol mode retrieval is possible. Such a profile would be rejected from the aerosol data set as cloud contaminated, but would also not count towards the cloud mode statistics. However, accounting for these low optical depth profiles would not be trivial. Aerosol mode retrievals can be made for aerosol optical depths of up to 5 to 7 (Giles *et al*, 2019), but there is no specific

corresponding threshold in cloud optical depth. In the interests of ensuring the profiles that could *potentially* be observed by AERONET in cloud mode are included, we choose to retain all low cloud optical depth profiles in the analysis, although recognise that the frequency of occurrence of such profiles is likely to be overestimated.

We begin by analysing profiles from SGP – a mid-latitude site whose cloud regimes consist of both frontal and convective

clouds with an overall average cloud fraction of about 50% (Lazarus *et al*, 2000). Ice fraction for SGP profiles is shown as a histogram in Figure 6a. Of the profiles, 26.3% contain cloud that is purely liquid and 33.3% contain cloud that is purely ice. Of the remaining 40.2% that contain mixed-phase cloud, profiles that are mostly liquid ($f < 0.5$) outnumber those that are mostly ice ($f > 0.5$) by about three to one.

Most of the profiles containing cloud that is either mostly or entirely ice have a low optical depth, and would therefore provide small contributions to long-term error statistics in a cloud optical depth climatology from AERONET (Figure 7a). Conversely, optical depth values for liquid or mostly liquid profiles tend to be greater, but the contributions to overall mean error are also likely to be small on account of low values of ice fraction. The contours on all panels of Figure 7 indicate the error that would result in an AERONET retrieval as a function of optical depth and ice fraction following Equation 1. At SGP, just under one

in ten of the profiles would have a cloud optical depth retrieval error of greater than 10 (9.2%), while only 3.1% of the profiles lie in the region where the error would be 20 or greater. The mean error across all profiles would be **3.5**.

At NSA, cloud fraction tends to be higher than SGP at about 75% (Dong *et al,* 2010), consisting of mostly stratiform cloud. There is a prevalence of thick, low-level mixed-phase cloud (Mülmenstädt *et al*, 2012), particularly in the summer when most

NSA profiles occur (note that NSA is inside the Arctic Circle, so no AERONET profiles are possible in the perpetual darkness of winter). Table 1 shows that there is a much greater frequency of mixed-phase clouds at NSA with respect to SGP, with much fewer profiles occurring that are either pure liquid or pure ice (Figure 6b). The result is a higher frequency of optically thicker clouds that are mostly ice, but a lower frequency of optically thicker profiles that are entirely ice (Figure 7b). The mean error in cloud optical depth as NSA is **4.4** – slightly higher than at SGP.


At the three tropical sites, the clouds tend to be much deeper and convective in nature, with a much greater occurrence of upper-level ice clouds (Stubenrauch *et al*, 2010). Despite their relative proximity, however, the meteorological conditions at the three sites are quite different. Manus is situated in the western Pacific "warm pool", and experiences much more convective activity throughout the year (Jakob and Tselioudis, 2003), while Nauru is on the edge of the warm pool and experiences much



less, although with a strong influence from the phase of the El Niño Southern Oscillation (Long *et al*, 2013). In contrast, Darwin experiences a strong seasonal cycle in its convective activity associated with the passage of the Australian Monsoon, with deep mixed-phase clouds occurring seasonally (Protat *et al*, 2011).

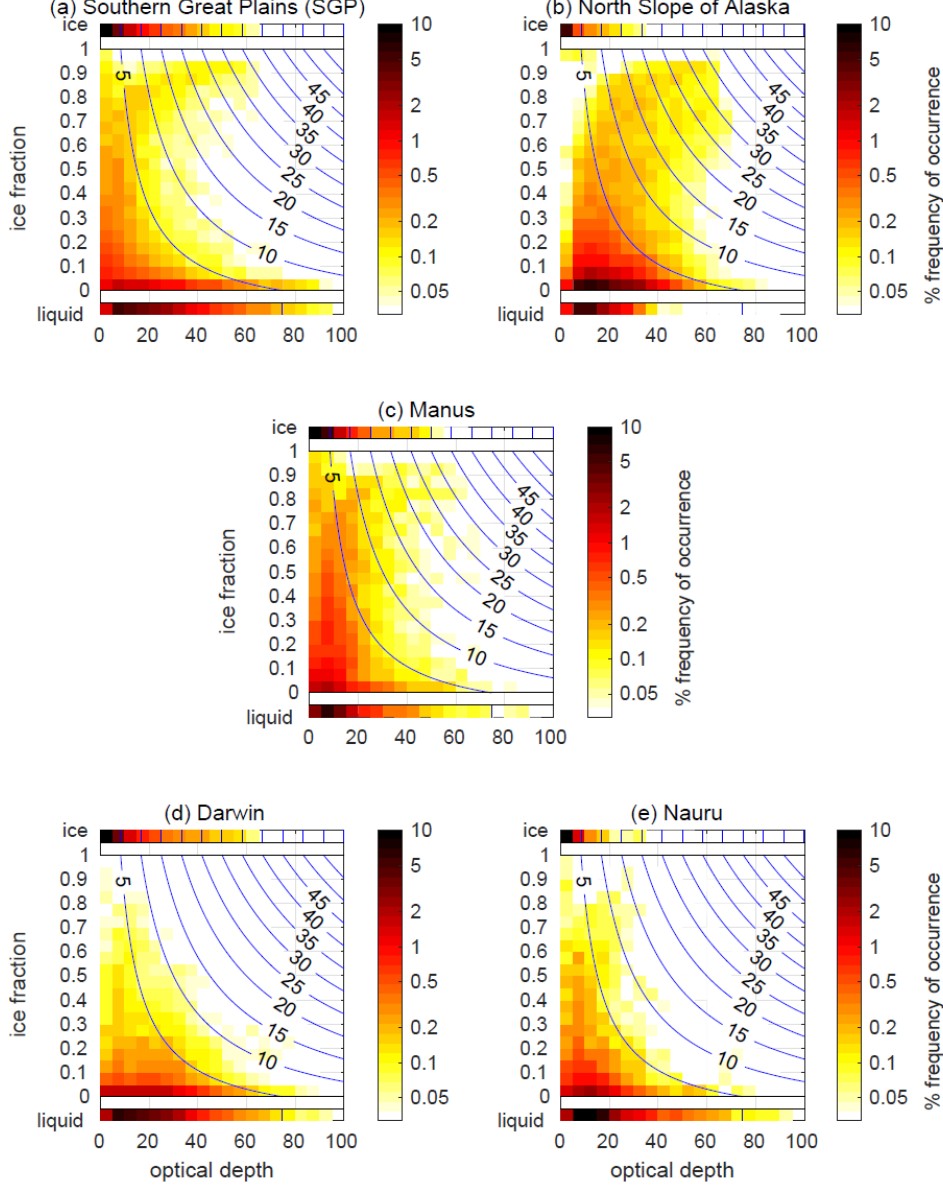

**Figure 7. Two-dimensional histograms of ice fraction and cloud optical depth at the five ARM sites for the same set of profiles as in Figure 6. The "liquid" and "ice" rows show the optical depth distribution of the profiles that contain purely ice or liquid; the rest of the plot separates the mixed-phase clouds by ice fraction as in Figure 6. The colour scale indicates the fraction of the total number**





**of profiles in each two-dimensional bin. The blue lines show the absolute error in retrieved optical depth that would result from AERONET retrievals as a function of ice fraction and cloud optical depth, calculated from Equation 1.**

The prevalence of deep convection at the three sites reflects the differences in frequency of profiles with high ice fraction (Figures 6c, 6d and 6e). The total frequency of mixed-phase profiles that have an ice fraction greater than 0.5 is 14.4% at

Manus, 4.8% at Darwin and 4.1% at Nauru. The greater frequency of convection at Manus appears as a higher fraction of profiles that are mixed-phase with high ice fractions (Figure 7c), resulting in the greatest overall error across the tropical sites (**3.5**). The much lower frequency of convection at Nauru results in fewer profiles appearing in this area of the histogram (Figure 7e), and hence a much smaller overall error (**1.8**). With an intermediate amount of convection and a greater fraction of optically thick ice cloud, the mean error at Darwin lies between the values at Manus and Nauru (**2.8**).

The analysis above from the five ARM sites implies that, if an estimate of ice fraction is not available at a given AERONET site, using uncorrected retrieved optical depths will lead to a mean error of order 2–4 in long-term statistics. Assuming typical mean cloud effective radius values of 6–12 µm, cloud optical depth errors of 2–4 are equivalent to errors in liquid water path of 8–32 g m$^{-2}$ (using Equation 2 in Chiu *et al*, 2012), which is of similar magnitude to retrieval uncertainty in liquid water path

from microwave radiometer observations (Marchand *et al*, 2003; Crewell and Löhnert, 2003).

To compare these uncertainties to a relevant climate variable, let us set out to retrieve cloud optical depths to sufficient accuracy that both top-of-atmosphere and surface fluxes are correct to within 10%. According Figure SB1 of Turner *et al* (2007), for a liquid cloud with a liquid water path of 100 g m$^{-2}$ and an effective radius of 8 µm, a typical top-of-atmosphere shortwave flux

would be 500 W m$^{-2}$, and the sensitivity of the top-of-atmosphere flux to the liquid water path about 1 W m$^{-2}$ (g m$^{-2}$)$^{-1}$. In this case, reproducing the top-of-atmosphere flux to within 50 W m$^{-2}$ implies a need for retrieval with an error of less than 50 g m$^{-2}$, equivalent to a cloud optical depth error of about 10. The mean AERONET cloud mode error of 2–4 is within this limit. By a similar argument, the presence of the same liquid cloud would result in a surface flux of about 300 W m$^{-2}$ with a sensitivity of surface flux of about 2 W m$^{-2}$ (g m$^{-2}$)$^{-1}$. To get the 10% accuracy in surface flux, the retrieval then would need to be accurate

to less than about 15 g m$^{-2}$ in liquid water path, or 3 in optical depth. Our errors may be slightly higher than this limit in some locations, and could only reach ~15% accuracy in surface flux.

Needless to say, if an independent estimate of ice fraction is available, we advocate the use of Equation 1 as a correction factor. Given that it is specific to the retrieval algorithm, it will be globally applicable to radiance measurements from any AERONET

radiometer under the assumption that the ice crystals in a cloud are rough, consist of a mixture of shapes and have effective diameters in the range 25 µm to 100 µm. At present, not all AERONET sites have the instrumentation to allow an ice fraction estimate to be made. A potential method to detect particle phase using AERONET radiometers that are polarimetrically sensitive could help with estimates of ice fraction, although further work is needed (Knobelspiesse *et al*, 2015). A weakness of Equation 1 is that it may not perform well at low optical depths. There are two possible solutions for this: first, via further





regressions, modifications to the equation could be made to add a component that describes the complicated dependencies in the "non-linear regime" at low optical depths, although the result would be invariably be a less simple equation. Second, alternative methods could be employed to retrieve optical depth in this range – for example, that of Hirsch *et al* (2012), although this would require the installation of specialised radiometers. Also, Guerrero-Rascado *et al* (2013) propose a method to obtain

cloud optical depth estimates using cloud-contaminated AERONET aerosol mode observations, which could provide an alternative source of data for low cloud optical depths.

## 5   Summary and conclusions

The representation of cloud properties in climate models still presents a huge challenge to climate scientists. To make progress in our understanding of cloud processes, we need global observations of cloud optical depth at high spatial and temporal

resolution. Ground-based measurements are best suited to provide such resolution, although global coverage is limited. The radiometers of the Aerosol Robotic Network (AERONET) could be readily used to increase the number of sites around the world by making routine "cloud-mode" measurements made during the "down time" when aerosol measurements are not possible. Retrievals are made using radiance at two wavelengths (440 nm and 870 nm) and a set of look-up tables. However, as the method is not able to retrieve cloud phase, the assumption is made that all of the retrieved optical depth is due to the

presence of warm, liquid cloud – hence, for any cloudy profile that contains an ice cloud component, there is likely to be an error in the retrieval.

We began by investigating the sign and magnitude of this "warm cloud error". A set of idealised cloud profiles were generated with varying total optical depth and "ice fraction" (the fraction of optical depth in the profile that is due to the presence of ice

cloud). We calculated the radiances that would be observed by a radiometer at the surface underneath the cloud profiles, and then used these radiances to retrieve the cloud optical depth. Comparison of the retrieved optical depths with the true, prescribed optical depths revealed that, for profiles that are mostly or entirely ice, the fractional error in retrieved optical depth was between 55% and 70% for ice particle diameters between 25 µm and 100 µm. At optical depths above 20, the fractional error was found to be a simple linear function of ice fraction and showed negligible dependence on optical depth or solar zenith

angle. Using a simple linear regression, we were able to generate an empirical equation (Equation 1 in this paper) linking the fractional error to the ice fraction. This equation has the potential to be used as a correction factor for AERONET optical depth retrievals. However, independent estimates of ice fraction are needed, which is not possible at most AERONET sites.

We then estimated the error in retrieved optical depth for a range of profiles of real clouds. We used multiple years of cloud

data from five sites of the Atmospheric Radiation Measurement (ARM) program, which were then sampled to include only profiles that could potentially be observed by an AERONET radiometer in cloud mode. Using Equation 1, an estimate of the retrieval error was generated for each profile. Clouds that were mostly ice tended to have lower optical depths, while optically

thicker clouds tended be mostly or entirely liquid – both of these conditions lead to small errors. At each of the five sites, only ~15% of the profiles had an error in retrieved cloud optical depth of larger than 10. The magnitude of the mean error at each location was dominated by the frequency of occurrence of optically thick clouds that were mostly or entirely ice – that is, either thick frontal cloud or deep convection. At the two sites located outside the tropics, where thick frontal cloud is the largest error

contribution, the overall mean error was related to the frequency of occurrence of such optically thick mixed-phase clouds. In the tropics, the error at each location was related to the frequency of occurrence of deep convection, with much greater variety in the error statistics. This suggests that variations in convective cloud occurrence may have a greater influence on the overall error than variations in frontal cloud occurrence.

The mean value of optical depth retrieval error at the five ARM sites is typically in the range 2 to 4. We showed that errors of this magnitude are small enough to allow the calculation of top-of-atmosphere fluxes to within 10% accuracy, and surface fluxes to within about 15%. Furthermore, when expressed in terms of liquid water path, these errors are of comparable value to uncertainties in retrievals from microwave radiometers. These results alone suggest that AERONET cloud mode retrievals could be a valuable source of cloud optical depth data from a large network of surface observation sites. A higher degree of

accuracy may be possible, though, via the use of a correction equation if an independent estimate of ice fraction can be obtained at the AERONET site.

**Data Availability**

The ARM Mace data used in this study can be accessed from www.arm.gov/data/data-sources/atmcldradmace-19. AERONET

cloud mode data can be accessed from aeronet.gsfc.nasa.gov/cgi-bin/type_piece_of_map_cloud.

**Author Contribution**

The experiment design and analysis was performed by JKPS and JCC. JKPS prepared the manuscript with contributions and advice from all co-authors.

**Competing Interests**

The authors declare that they have no conflict of interest.





**Acknowledgements**

This work was supported by the Office of Science (BER), DOE under grants DE-SC0011666, DE-SC0018930 and DE-SC0018045. We thank the AERONET team for calibrating and maintaining instrumentation and processing these data.

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
