# Peer review of "The Impact of Neglecting Ice Phase on Cloud Optical Depth Retrievals from AERONET Cloud Mode Observations"

_Atmospheric Measurement Techniques, 2019_

## Referee Comment (RC1) · Darrel Baumgardner (Referee) · 12 Jun 2019

Kudos to the authors for this very well executed study and write-up. Implementation of the suggested linear correction factor for measured optical depths larger than 20 will provide a much needed improvement to current data bases of cloud optical depth and fraction.

The only question that I have for the authors, who are free to address as they wish, is if local sounding information could be used to constrain estimates of cloud ice fraction when other types of measurements from radar or microwave radiometers are not available. Estimates of cloud base/top temperatures that can be derived from the 0 or 12Z

soundings might be useful in this regard.

---

## Referee Comment (RC2) · Anonymous Referee #3 · 29 Jun 2019

Summary: This paper reports the results of an algorithm that builds naturally on earlier ones developed by Marshak, Chiu, and others for retrieving cloud optical depth from two-channel (visible and near infrared) downwelling radiance measurements. Here, the goal of the algorithm enhancements is to reduce retrieval error when ice particles, as well as liquid particles, contribute to the downwelling radiances. The paper is easy to understand and all of the analyses make sense. I recommend its publication in AMT. My comments are minor in nature.

Comments:

1) The algorithm makes use of one-dimensional radiative transfer theory and differen-

tial surface reflectivity at the two wavelengths to compensate for it. After reading the Introduction it was not clear how accurate this approach is when applied to different cloud types. In particular, how good is it when applied to fields of horizontally small but vertically developed convective clouds in the tropics as a function of overall cloud fraction? Does significant side illumination and/or cloud side leakage of the convective elements cause problems? If Aeronet is located near a coastal site, do differences in the surface reflectivity of the land and the ocean cause problems? These types of questions are relevant to the paper because the algorithm is applied to tropical convective clouds at sites with mixtures of surface types in their environs. Strengthen the Introduction by making clear, with supporting references, how accurate the retrieval is expected to be for different cloud types and mixtures of different surface types. If this is not known, then say so and caveat your optical depth retrieval accuracies towards the end of the paper. A study that uses three-dimensional radiative transfer theory to pound on these types of uncertainties would be valuable if such a study has yet to be performed. If it has, do reference and discuss it within the Introduction.

2) Lines 23-25 on Page 5 and Lines 1-2 on Page 6 indicate that the radiative transfer is always performed with the ice in a top layer and the liquid in a bottom layer. So these calculations are done with ice on top and liquid underneath and not for what are traditionally called mixed-phase clouds. Does it make any difference if the layers are mixed up together to form what is generally called a mixed-phase cloud? On Page 10, Line 4, the paper refers to mixed-phase cloud in the traditional sense. But on Page 12, Line 12, it is not clear what mixed phase means here. Just be sure to be clear everywhere exactly how the liquid and ice are being dealt with. It may not make any difference for the calculations, but it sure does make a difference for the retrievals: retrieving the properties of liquid only and ice only clouds is not easy but it sure is easier than retrieving the properties of ice and liquid particles all mixed together. Error estimates for the retrieved liquid and ice particle properties would be more convincing if they were provided in the context of the types of clouds above. These context-based error estimates would be of value if percolated into uncertainties for the retrieved optical

depths. For example, in convective clouds with mixtures of in-cloud rain, in-cloud ice precipitation, liquid cloud, and ice cloud, not clear at all as to what the actual errors in the retrievals might be.

Minor Details:

0) A marked-up manuscript is being returned to the authors; perhaps some of the mark up may be of value to them.

1) First sentence of the abstract: "Cloud optical depth remains a difficult variable to represent in climate models" might be true for a bunch of different reasons not related to "a need for high-quality observations of cloud optical depth from locations around the world". So, the first sentence of the abstract is not compelling.

2) The words "could", "could be", "can be",..., are used a lot in the paper. These are weak words in a scientific context and replacing them all with well thought out stronger words would improve the paper.

3) Page 4, Figure 1: The dashed line is really hard to see close to 0.

4) Page 5, Line 13: Wrong units for radiance.

5) Page 6, Figure 3: Why not squares for the top row of figures with the same x- and y-axis range? A line along the diagonal would help too.

6) Page 8, Lines 23-24: "hence far less of an issue ..." is a subjective statement and would depend upon the application. As such, it is not a correct statement for all situations.

7) Page 10, Lines 14-19: Past tense would probably be better for describing what you did to execute the study.

8) Page 13, Figure 7: Make sure all of the minor tick marks show up in the figure.

Please also note the supplement to this comment:

https://www.atmos-meas-tech-discuss.net/amt-2019-169/amt-2019-169-RC2-supplement.pdf

Interactive
comment

---

## Author Comment (AC1) · 20 Aug 2019

We thank both reviewers for their suggestions and comments. Our responses are presented below. A marked-up version of the manuscript has been uploaded with changes highlighted.

Reviewer "A" Kudos to the authors for this very well executed study and write-up. Implementation of the suggested linear correction factor for measured optical depths larger than 20 will provide a much needed improvement to current data bases of cloud optical depth and fraction.

1. The only question that I have for the authors, who are free to address as they wish, is if local sounding information could be used to constrain estimates of cloud ice fraction when other types of measurements from radar or microwave radiometers are not available. Estimates of cloud base/top temperatures that can be derived from the 0 or 12Z soundings might be useful in this regard.

—> To apply the correction equation to AERONET cloud mode retrievals, we ideally need instantaneous estimates of ice fraction that could be aligned with an instantaneous AERONET retrieval of optical depth. However, approaches using radiosonde soundings (or also satellites) could be used to provide a general estimate of ice fraction over an area and a period of time. However, if the cloud regime changes, or the cloud formations are varying rapidly, this could be a source of extra error. Extra work would be needed on this. We have added sentences about this in the Discussion section (Section 5; page 15, lines 18—22).

---

## Author Comment (AC2) · 20 Aug 2019

We thank both reviewers for their suggestions and comments. Our responses are presented below. A marked-up version of the manuscript has been uploaded with changes highlighted.

Reviewer "B" Summary: This paper reports the results of an algorithm that builds naturally on earlier ones developed by Marshak, Chiu, and others for retrieving cloud optical depth from two-channel (visible and near infrared) downwelling radiance measurements. Here, the goal of the algorithm enhancements is to reduce retrieval error when ice particles, as well as liquid particles, contribute to the downwelling radiances.

The paper is easy to understand and all of the analyses make sense. I recommend its publication in AMT. My comments are minor in nature.

Comments: 1. The algorithm makes use of one-dimensional radiative transfer theory and differential surface reflectivity at the two wavelengths to compensate for it. After reading the Introduction it was not clear how accurate this approach is when applied to different cloud types. In particular, how good is it when applied to fields of horizontally small but vertically developed convective clouds in the tropics as a function of overall cloud fraction? Does significant side illumination and/or cloud side leakage of the convective elements cause problems? If Aeronet is located near a coastal site, do differences in the surface reflectivity of the land and the ocean cause problems? These types of questions are relevant to the paper because the algorithm is applied to tropical convective clouds at sites with mixtures of surface types in their environs. Strengthen the Introduction by making clear, with supporting references, how accurate the retrieval is expected to be for different cloud types and mixtures of different surface types. If this is not known, then say so and caveat your optical depth retrieval accuracies towards the end of the paper.

-> The performance of the method was examined by Marshak et al (2004) and Chiu et al. (2006), who showed that the method works well for both overcast and broken cloud fields. The method will not work well when clouds do not fully cover the field of view (FOV) of the radiometer (the so-called "clear-sky contamination" issue; see Chiu et al., 2006). Therefore, unphysical cloud optical depth can happen near cloud edges. Such contamination is more frequent in small cumulus clouds, although convective clouds have sufficiently large horizontal extents that they can completely cover the narrow 1.2° FOV of the sun-photometers. Note that when a time series of retrievals is available (e.g., cases in Chiu et al., 2006), one can detect unphysical retrievals near cloud edges and remove them. For AERONET cloud mode retrievals which are not made from a complete time series, it is more difficult to detect these unphysical retrievals. For this reason, AERONET reports a "cluster" average (see Chiu et al., 2010), excluding
retrievals below the 25th and above the 50th percentile – a similar approach to that of Remer et al (2005), for aerosol retrievals.

-> Differences in surface reflectivity is important to consider for cloud mode retrievals. Surface albedo information is considered over a  $4 \times 4$  km domain surrounding the AERONET site, and a combination of land and ocean surfaces surrounding a site is not ideal. For this reason, the sites included in the AERONET cloud mode dataset have been selected to ensure that the spectral contrast from surrounding vegetated surface is sufficient for the retrieval method (see Chiu et al, 2010).

-> We have added more detail addressing these questions into the fourth paragraph of the Introduction section and divided the paragraph into two (see page 3, lines 3—15).

A study that uses three-dimensional radiative transfer theory to pound on these types of uncertainties would be valuable if such a study has yet to be performed. If it has, do reference and discuss it within the Introduction.

-> The reviewer is right; such studies would be valuable. While a lot of studies have focussed on reflected radiance at the top of the atmosphere, there is actually no published paper for zenith radiance at the surface. We looked into 3D radiative effects on cloud mode retrievals a long time ago, but have not found time yet to wrap this up. We incorporate the reviewer's point in the manuscript and, hopefully, this could also motivate others to conduct thorough analyses on 3D radiative transfer.

2. Lines 23-25 on Page 5 and Lines 1-2 on Page 6 indicate that the radiative transfer is always performed with the ice in a top layer and the liquid in a bottom layer. So these calculations are done with ice on top and liquid underneath and not for what are traditionally called mixed-phase clouds. Does it make any difference if the layers are mixed up together to form what is generally called a mixed-phase cloud?

 $\rightarrow$  Mixing up the ice and liquid will affect the radiance (see Sun and Shine, 1994) – the zenith radiance will be slightly higher if the ice and liquid particles are fully mixed
rather than in two separate layers as in our idealised calculations. More work would be required to understand and quantify the effect of this "mixing" on the correction equation. We have included this question in the future work paragraphs at the end of Section 5 (page 16, lines  $1\hat{a}\check{A}\check{T}$ 7).

On Page 10, Line 4, the paper refers to mixed-phase cloud in the traditional sense. But on Page 12, Line 12, it is not clear what mixed phase means here. Just be sure to be clear everywhere exactly how the liquid and ice are being dealt with. It may not make any difference for the calculations, but it sure does make a difference for the retrievals: retrieving the properties of liquid only and ice only clouds is not easy but it sure is easier than retrieving the properties of ice and liquid particles all mixed together.

-> We have been quite inconsistent with what we mean by "mixed-phase" clouds in this study, especially in Section 4. We have now tidied it up in various places so that "mixed-phase" is only used to mean mixtures of ice and liquid particles, while general clouds containing both liquid and ice have been described as such. It should now be clearer in both places mentioned in the comment above. We have also cleared it up in Table 2, where the words "mixed-phase cloud" have been replaced with "ice and liquid cloud".

Error estimates for the retrieved liquid and ice particle properties would be more convincing if they were provided in the context of the types of clouds above. These contextbased error estimates would be of value if percolated into uncertainties for the retrieved optical paper depths. For example, in convective clouds with mixtures of in-cloud rain, in-cloud ice precipitation, liquid cloud, and ice cloud, not clear at all as to what the actual errors in the retrievals might be.

-> We thank the reviewer for this suggestion. We agree that it would be good to provide errors in the context of cloud types, but we wish to leave it for future work because it requires substantial work and proper ancillary datasets to address this issue properly. Note that AERONET sun-photometers only operate in the absence of precipitation to
keep lenses clean and dry. Therefore, we actually do not have many observations for convective clouds described above. If in-cloud rain evaporates before reaching the ground, we have chances to sample such clouds and provide retrieval. However, we have found rain (or drizzle in most cases) do not affect zenith radiance significantly due to its small number of concentrations, based on work in Fielding et al. (2014, 2015). It would be important to tackle the properties of precipitating ice particles, but current retrievals and model simulations for ice microphysical and optical properties are quite uncertain. We are currently working on ice retrieval using polarimetric radar measurements. Hopefully, we can find good collocated datasets to address this.

Minor Details: A marked-up manuscript is being returned to the authors; perhaps some of the mark up may be of value to them.

-> Thank you – this was a helpful inclusion. We have made a number of modifications following your suggestions on the mark-up version (not all of which are mentioned directly in a comment here).

3. First sentence of the abstract: "Cloud optical depth remains a difficult variable to represent in climate models" might be true for a bunch of different reasons not related to "a need for high-quality observations of cloud optical depth from locations around the world". So, the first sentence of the abstract is not compelling.

-> We have reworked the first few sentences of both the Abstract and the Introduction to better link the challenges associated with modelling clouds to the need for cloud optical depth observations. Also, we have reworked and edited the Abstract following the many comments on Reviewer B's marked-up manuscript.

4. The words "could", "could be", "can be",..., are used a lot in the paper. These are weak words in a scientific context and replacing them all with well thought out stronger words would improve the paper.

-> All instances of these words (as highlighted in the reviewer's marked-up document)
have been reviewed and, where appropriate, stronger words have been used.

5. Page 4, Figure 1: The dashed line is really hard to see close to 0.

-> The lines on Figure 1 are now coloured to make the contrast easier. For consistency, the colouring of the lines has been changed in Figure 2 to match.

6. Page 5, Line 13: Wrong units for radiance.

-> The units for radiance both here (now page 6, line 3) and in the y-axis label of Figure 2 should actually be dimensionless, as they correspond to radiances calculated for a unit flux at the top of the atmosphere. This has been corrected, and the normalised nature of the radiances explained in the caption in Figure 2. We have also clarified that the radiances presented here are normalised in the text (page 5, lines 18—19).

7. Page 6, Figure 3: Why not squares for the top row of figures with the same x- and y-axis range? A line along the diagonal would help too.

-> The square axes for the top panels of Figure 3 is not practical, as extending the range of true optical depths to 100 would result in the inclusion of many retrievals that exceed 100 (the maximum optical depth in the AERONET look-up tables; this limit has now been mentioned in Section 2). Hence we have only included true optical depths up to 50 as, in this range, none of the retrievals (however high their ice fraction) exceed 100. For clarity, however, we have added the one-to-one line on the top row of panels in Figure 3 as requested. For consistency, we have also added zero lines on the bottom row of Figure 3 and all of Figure 4.

8. Page 8, Lines 23-24: "hence far less of an issue ..." is a subjective statement and would depend upon the application. As such, it is not a correct statement for all situations.

-> Our paper centres on assessing whether the errors affect the long-term cloud optical depth statistics, but we do recognise that there may be instances where accurate retrievals are required at low optical depths. Greater accuracy at low optical depths AMTD
could be achieved by generating an improved, more complex version of Equation 1. We have highlighted this in Section 3 (page 9, lines 14—18) and then again in a little more detail in Section 5 (page 15, lines 28—30). This Discussion section is a new addition that brings together the various discussion points from the results sections in one place.

9. Page 10, Lines 14-19: Past tense would probably be better for describing what you did to execute the study.

-> The paragraphs describing the data and how we sampled it is all now in the past.

10. Page 13, Figure 7: Make sure all of the minor tick marks show up in the figure.

-> The grid on Figure 7, which previously was drawn behind the coloured boxes of the 2D histogram, is now replotted on top (but under the blue contours).

---

## Author Response (AR1)

**The Impact of Neglecting Ice Phase on Cloud Optical Depth Retrievals from AERONET Cloud Mode Observations**

Jonathan K. P. Shonk1, Jui-Yuan Christine Chiu2, Alexander Marshak3, David M. Giles3, 4, Chiung-Huei 5 Huang5, Gerald G. Mace6, Sally Benson6, Ilya Slutsker3, 4 and Brent N. Holben3

1National Centre for Atmospheric Science, Department of Meteorology, University of Reading, Reading, UK
 2Department of Atmospheric Science, Colorado State University, Fort Collins, CO, 80523, USA
 3NASA/Goddard Space Flight Center, Greenbelt, Maryland, USA
 4Science Systems and Applications, Inc., Lanham, Maryland, USA

5Center for Environmental Monitoring and Technology, National Central University, Taoyuan, Taiwan
 6Department of Atmospheric Sciences, University of Utah, Salt Lake City, Utah, USA

Correspondence to: Jonathan K. P. Shonk (j.k.p.shonk@reading.ac.uk)

Abstract. Clouds present many challenges to climate modelling. To develop and verify the parameterisations needed to allow climate models to represent cloud structure and processes, there is a need for high-quality observations of cloud optical depth

- 15 from locations around the world. Retrievals of cloud optical depth are obtainable from radiances measured by Aerosol Robotic Network (AERONET) radiometers in "cloud mode" using a two-wavelength retrieval method. However, the method is unable to detect cloud phase, hence assumes that all of the cloud in a profile is liquid. This assumption has the potential to introduce errors into long-term statistics of retrieved optical depth for clouds that also contain ice. Using a set of idealised cloud profiles we find that, for optical depths above 20, the fractional error in retrieved optical depth is a linear function of the fraction of the
- 20 optical depth that is due to the presence of ice cloud ("ice fraction"). Clouds that are entirely ice have positive errors with magnitudes of order 55% to 70%. We derive a simple linear equation that can be used as a correction at AERONET sites where ice fraction can be independently estimated.

Using this linear equation, we estimate the magnitude of the error for a set of cloud profiles from five sites of the Atmospheric

25 Radiation Measurement programme. The dataset contains separate retrievals of ice and liquid retrievals, hence ice fraction can be estimated. The magnitude of the error at each location was related to the relative frequencies of occurrence in thick frontal cloud at the mid-latitude sites and of deep convection at the tropical sites; that is, of deep cloud containing both ice and liquid particles. The long-term mean optical depth error at the five locations spans the range 2–4, which we show to be small enough to allow calculation of top-of-atmosphere flux to within 10%, and surface flux to about 15%.

30

**Commented [JS1]:** B3. The opening sentence of the Abstract has been reworded so that the motivation reads better. Much of the Abstract has also been reordered and edited following many comments on the mark-up document from Reviewer B.

**1 Introduction**

5

Clouds are a crucial part of the climate system, yet present many great challenges to climate science (Randall *et al*, 2007; Boucher *et al*, 2013). Despite recent progress, climate models struggle to represent the optical properties of clouds (Bender *et al*, 2006; Lauer and Hamilton, 2013; Klein *et al*, 2013; Calisto *et al*, 2014). Cloud optical depth is particularly important to represent reliably as it governs the effect of clouds on the Earth's radiation budget. The complex processes and interactions

- that describe the evolution of clouds occur on scales much smaller than a model grid box and hence require parameterisation (Pincus *et al*, 2003; Shonk and Hogan, 2010). To develop and validate these parameterisations, there is a need for global observations of cloud optical depth at high temporal and spatial resolution.
- 10 A common approach to measure cloud optical depth is to retrieve it remotely from measurements of reflectance, radiance or irradiance in multiple spectral bands. Various methods have been developed to retrieve cloud optical depth from satellite measurements (for example, Arking and Childs, 1985; Nakajima and King, 1990; Platnick *et al*, 2001; Cooper *et al*, 2007) and ground-based instruments (Marshak *et al*, 2000, 2004; Barker and Marshak, 2001; Chiu *et al*, 2006). The need for global observations is best met by satellites, which are capable of providing routine cloud optical depth retrievals all around the world.
- 15 However, on account of their large pixel size, they struggle to provide the high temporal and spatial resolution required to investigate cloud processes. The underlying surface adds to the complexity of variability in the optical properties, and broken clouds and subpixel clouds increase the chance of errors and biases (Stephens and Kummerow, 2007). Using ground-based observations eliminates many of these issues. The proximity of clouds to the ground (much closer than a satellite orbit) means that a radiometer can achieve much smaller pixel sizes for the same viewing angle, allowing much higher temporal and spatial 20 resolution, and reducing the incidences of cloud edge.

A disadvantage of using ground-based observations is the lack of global coverage. We are limited to the small number of locations around the world where routine cloud optical depth observations are made: until recently, sites of the Atmospheric Radiation Measurement (ARM) Programme (Stokes and Schwartz, 1994) and the sites of the Aerosols, Clouds and Trace

- 25 Gases Research Infrastructure (ACTRIS) network that were formerly part of Cloudnet (Illingworth et al, 2007). But Chiu et al (2010) noted that radiometers distributed throughout the world as part of the AERONET project (Holben et al, 1998) could provide a readily available source of cloud optical depth observations and hence provide greater global coverage. When the sun is not obscured by cloud, these radiometers are in "aerosol mode" and make regular measurements of aerosol properties. When the sun is obscured, however, aerosol measurements are not possible and the radiometer becomes idle. Marshak et al
- 30 (2004) proposed that the "down-time" when the aerosol measurements are not possible could be used to observe cloud properties ("cloud mode") via measurements of zenith radiance.

**Commented [JS2]: B3.** These sentences have been reordered and edited similarly to the start of the Abstract.

**Commented [JS3]: B.**

Commented [JS4]: B.

Cloud optical depth retrievals are made using the method proposed by Chiu et al (2010). It is based on that of Marshak et al (2004), and uses zenith radiances measured at two wavelengths (440 nm and 870 nm; one visible, one infra-red) to retrieve cloud optical depth and cloud fraction. Above a green, vegetated surface, the radiative properties of the clouds are similar at these wavelengths, but there is a strong contrast in surface albedo. Retrieval is performed using a set of radiance look-up tables

5 calculated at the two wavelengths. The approach has been shown to be applicable for both overcast and broken cloud fields (Chiu et al, 2006), and performed well when applied to an artificial field of clouds whose optical depth was known (Marshak et al, 2004). A limitation to the method is that it does not perform well near cloud edge: clear-sky contamination of the field of view, and high radiances arising from direct solar illumination of cloud edge, can both generate unrealistic optical depths (Chiu et al, 2006). In AERONET, contamination problems are reduced by clustering retrievals into 1.5-minute intervals and 10 excluding extreme optical depth values (Chiu et al, 2010).

Using this method, AERONET "cloud mode" optical depth retrievals have now been made routinely at a number of sites around the world for several years. A requirement for a "cloud mode" site is that the surrounding area is generally green vegetation: suitable AERONET sites were selected using satellite-derived contrasts in albedo at the two wavelengths (Chiu et

- 15 al, 2010). Cloud mode retrievals from AERONET are beginning to appear in published studies. An evaluation of data from one AERONET site in Cuba was made by Barja et al (2012). Antón et al (2012) used cloud mode data in a study into the effects of cloud optical depth on the transmission of ultra-violet radiation; Li et al (2018) used it to investigate seasonal and spatial distributions of cloud optical depth across China alongside satellite optical depth retrievals from MODIS (the Moderate Resolution Imaging Spectroradiometer; Platnick et al, 2003). An AERONET radiometer was also taken aboard a ship to probe the properties of boundary layer cloud in the north-eastern tropical Pacific (Painemal et al, 2017). 20

An extension to the retrieval method by Chiu et al (2012) included a third wavelength (1640 nm), which allows a retrieval of cloud droplet effective radius to be obtained alongside cloud optical depth and cloud fraction. Effective radius retrievals tend to be very sensitive to uncertainty in surface albedo and radiance measurements, so Chiu et al (2012) suggested performing

- the retrieval 40 times with perturbations to surface albedo and the measured radiance, thereby providing mean values of the 25 retrieved values and an estimate of the uncertainty in these retrievals. This method was used in the study of Painemal et al (2017), although the standard retrievals available on the AERONET website use the two-wavelength method of Chiu et al (2010).
- 30 However, neither of these retrieval methods are capable of retrieving cloud phase, so an assumption is made. Given the tendency for the liquid component of a cloudy profile to be substantially optically thicker than the ice component, it is assumed that the entirety of the retrieved cloud optical depth value is due to the presence of liquid cloud. This "warm cloud assumption" has the potential, therefore, to introduce an error into cloud optical depth retrievals in any case where a cloudy profile contains ice cloud, which could cause problems in studies that analyse long-term statistics of cloud optical depth.

Commented [JS5]: B1. Some more information added about the performance of the retrieval method, including information about its performance under certain situations.

Commented [JS6]: B1. Discussion that cloud mode sites need to be generally surrounded by green vegetation.

[revised manuscript text omitted]

**Commented [JS9]: B5.** The lines have been coloured to make the dashed line clearer.

**Commented [JS12]: B6.** Text updated to indicate that the radiance is normalised.

Commented [JS11]: B5. Colours updated in the caption.

cloud that is either purely ice or purely liquid, for a prescribed solar zenith angle of 30° and a top-of-atmosphere flux of unity (hence the radiances presented are normalised). For a given optical depth, the observed radiance for liquid clouds is always
more than that for an ice cloud of the same optical depth over the entire range of effective sizes used in this study. This is because liquid droplets have a greater tendency to forward scatter than ice crystals, resulting in a greater radiance at the surface

Figure 2. Normalised radiances extracted from the liquid (red) and ice (blue and green) look-up tables for a range of different optical

depths, all calculated with unit top-of-atmosphere flux, for a solar zenith angle of 30° and at the visible 440 nm wavelength over a

[revised manuscript text omitted]